# Enhancing Value and Uptake for Whole-Population Cohorts of Children and Parents: Methods to Integrate Registries into the Generation Victoria Cohort

**DOI:** 10.3390/children8040285

**Published:** 2021-04-07

**Authors:** Valerie Sung, Katrina Williams, Ella Perlow, Yanhong J. Hu, Susannah Ahern, Joanne M. Said, Bill Karanatsios, John L. Hopper, John J. McNeil, Leo Donnan, Sharon Goldfeld, Melissa Wake

**Affiliations:** 1Murdoch Children’s Research Institute, Melbourne 3052, Australia; valerie.sung@rch.org.au (V.S.); katrina.williams@monash.edu (K.W.); ella.perlow@mcri.edu.au (E.P.); jessika.hu@mcri.edu.au (Y.J.H.); leo.donnan@rch.org.au (L.D.); sharon.goldfeld@mcri.edu.au (S.G.); 2Department of Paediatrics, The University of Melbourne, Melbourne 3052, Australia; 3The Royal Children’s Hospital, Melbourne 3052, Australia; 4Department of Paediatrics, Monash University, Melbourne 3168, Australia; 5Monash Children’s Hospital, Melbourne 3168, Australia; 6Department of Epidemiology and Preventive Medicine, Monash University, Melbourne 3004, Australia; susannah.ahern@monash.edu (S.A.); john.mcneil@monash.edu (J.J.M.); 7Department of Obstetrics and Gynecology, The University of Melbourne, Melbourne 3052, Australia; jsaid@unimelb.edu.au; 8Maternal Fetal Medicine, Joan Kirner Women’s & Children’s at Sunshine Hospital, Western Health, Melbourne 3021, Australia; 9Western Health Chronic Disease Alliance, Western Health, Melbourne 3021, Australia; Bill.Karanatsios@wh.org.au; 10Melbourne School of Population of Global Health, The University of Melbourne, Melbourne 3053, Australia; j.hopper@unimelb.edu.au; 11Liggins Institute, University of Auckland, Auckland 1023, New Zealand

**Keywords:** research methodology, registries, registry trials, population studies, GenV (Generation Victoria), children

## Abstract

Health registries are critical to understanding, benchmarking and improving quality of care for specific diseases and conditions, but face hurdles including funding, bias towards clinical rather than population samples, lack of pre-morbid and outcomes data, and absent cross-registry harmonisation and coordination. Children are particularly under-represented in registry research. This paper lays out novel principles, methods and governance to integrate diverse registries within or alongside a planned children’s mega-cohort to rapidly generate translatable evidence. GenV (Generation Victoria) will approach for recruitment parents of all newborns (estimated 150,000) over two years from mid-2021 in the state of Victoria (population 6.5 million), Australia. Its sample size and population denominator mean it will contain almost all children with uncommon or co-morbid conditions as they emerge over time. By design, it will include linked datasets, biosamples (including from pregnancy), phenotypes and participant-reported measures, all of which will span pre-morbid to long-term outcomes. We provide a vignette of a planned new registry for high-risk pregnancies to illustrate the possibilities. To our knowledge, this is the first paper to describe such a methodology designed prospectively to enhance both the clinical relevance of a large multipurpose cohort and the value and inclusivity of registries in a population.

## 1. Introduction

Health registries are critical to understanding disease prevalence, phenotype, prevention, progression and outcomes [1,2]. They provide cost-effective strategies for measuring and benchmarking real-world treatment outcomes to improve quality of care [2]. However, registries face many hurdles, and for children are lacking for many conditions and typically miss many milder cases managed in the community. The Dutch PHARMO Perinatal Research Network [3] and the Nordic registry programs [4] show that whole-population cohorts created from administrative datasets can obviate burden and be very complete, although lack important phenotypic, child and parent-reported, and biologic measures. On the other hand, previous large-scale (*n* = 80,000+) cohorts that include deep individual and cross-generational biological and phenotypic measures, such as the UK Life Study and the US National Children’s Study [5], suffered from very low uptake despite years of planning and funding. Burden of participation appeared to be a major factor.

One exciting solution might be a hybrid approach that combines linked and participant-collected data and biosamples into a single powerful, low-burden, consented longitudinal framework that also systematically integrates with health registries. Large cohorts could be designed to “stop describing and start fixing” [6], collaborating with multiple registries to provide longevity, comparable outcomes across conditions, design flexibility and support for registry trials. Cohorts could work with existing registries, enable new registries and/or conduct ‘depth’ studies. For the latter, if the cohort is sufficiently large with a whole-population denominator, it could essentially operate as multiple time-limited registries.

While birth cohorts can and do embed ‘focus’ or ‘depth studies’, these are typically small, arise from a local sample rather than population denominator, and reflect the targeted uptake into the overarching study. The ORIGINS study [7], a regional Western Australian birth cohort, supports a number of substudies created to answer specific research questions. The Norwegian Mother, Father and Child Cohort Study [8]—one of very few ‘mega’ birth cohorts worldwide—is linked to registries (such as the Cerebral Palsy Register in Norway) and supports depth studies for specific childhood conditions like Attention Deficit Hyperactivity Disorder (ADHD) and autism, although its 40% uptake limits its core registry capabilities. We have not found examples of very large cohorts designed from the outset to enhance practical value and return on investment by systematically embedding clinical registry research.

This paper describes what we believe is a novel, internationally unique methodology to bring significant advances to clinical care and epidemiology. Our design for a new whole-population child and parent cohort, GenV (Generation Victoria), is planned to maximise the benefits of registry research. We report on the developing processes and methods to enable this and provide an example of a new registry under development.

## 2. Materials and Methods

### 2.1. About GenV

GenV is a cohort in advanced planning that from 2021 will approach for recruitment parents of all newborns (estimated 160,000) over two years in the state of Victoria (population 6.5 million in 2019), Australia. Its primary objective is to create very large birth and parallel parent cohorts for observational and interventional research that benefits human health and wellbeing. While still evolving, GenV has been approved by The Royal Children’s Hospital Human Research Ethics Committee (HREC 2019.011), including in-principle approval as a mechanism to support registries.

Setting & design: Victoria’s relevant demography is summarised elsewhere [9]; >60% of Australian parents report their child has at least one ongoing health or developmental problem at any given time from age two, rising to >70% from age eight to 15 years [10]. GenV’s longitudinal cohort design comprises four building blocks: (1) consent soon after birth (with GenV recruiters based in every Victorian birthing hospital for two years), (2) retrospective and prospective linkage to clinical and administrative datasets, (3) banking of universal and clinical biosamples (including pregnancy), and (4) prospective GenV-collected data.

### 2.2. Principles and Concepts for GenV-Registry Collaboration

Clinical registries may be established primarily for epidemiological purposes, particularly for rare diseases and quality benchmarking [2], or research purposes including access to clinical trials [11]. All of these registry purposes can be supported in GenV. The mechanisms through which GenV and registries can integrate include data linkage, patient recruitment, and streamlining of data collection. GenV’s six principles are collaboration, inclusivity, sustainability, enhancement, systematised processes, and value. Figure 1 shows the six additional principles via which GenV and registries can integrate.

Figure 2 shows conceptually how registries for different conditions might intersect with GenV either within (Model 1) or alongside (Model 2) the cohort, while Figure 3 indicates a process flow for variations on these models [12]. Model 1 registries will start as depth sub-cohorts within GenV, so may be fully covered by its consent for children born in GenV’s birth window. Model 2 registries recruit their own participants (some facilitated via GenV opt-out or opt-in processes to pass on personal information), including consent to also participate in GenV to share data.

### 2.3. Preparatory Work

Content: We anticipate many ‘informal’ Model 1 registries whereby GenV data identifies that a child has an issue/condition (‘depth’ subcohorts), triggering additional condition-specific consent, within existing consent. More formal start-up Model 1 registries will likely work towards continuation beyond the GenV sample, so that recruitment sustainability will underpin establishment content. Existing registries and GenV will decide mutual benefit and feasibility of processes, in line with the principles and approaches described above; such conversations are already occurring (see Section 3). We envisage activities such as annual open face-to-face and web-based fora to brainstorm and prioritise registry ideas, ideally involving a range of stakeholders including policy, services, communities, and families.

Working together: Good communication, transparency and agreement are vital and will underpin a Working Together Agreement between GenV and each registry [13]. While GenV is seeking dedicated funds to provide some support capacity, registries will require their own funding for data sharing and other collaborative activities. Expertise and advice may also be sought from centres of registry expertise.

Agreement to proceed with a registry-GenV collaboration: Initial registry-GenV discussions will articulate the rationale of the existing or planned registry, its design and justification of its purpose, use and scope. If feasibility and mutual alignment appear likely, the registry would proceed to a partnering agreement (e.g., Data Sharing or Patient Recruitment) that defines at least the following 8 items: (1) the GenV registry model being followed; (2) design and high-level (or draft) protocol, taking participant burden into account; (3) timelines; (4) data sharing/linkage and governance plans; (5) status of ethical approval; (6) communication with participants, including information statement and consent; (7) registry oversight and quality processes; and (8) capacity assessment for registry continuation, including human resource and funding.

An Expression of Interest form will be available on GenV’s website. We are developing transparent process flows, to be overseen by a GenV Registry Oversight Committee with cross-disciplinary expertise and consumer involvement. The registry data custodian will be the sponsor, most likely from a university, research institute or similar organisation, but GenV does not preclude commercial sponsors provided its principles are met.

### 2.4. Consent

At recruitment into GenV, parents provide consent for approved researchers to access GenV’s data, data sharing between GenV and external registries, and recontact to offer additional research opportunities [14]. Consent wording for registries will vary by model (Figure 3). For Model 1, GenV already includes consent for data sharing with approved users. For Model 2, data will optimally be able to flow bidirectionally (from registry to GenV and vice versa) as outlined in Section 2.6 below. GenV will ensure contact with participant families about relevant new registries will be kept to a minimum and preferably at the same time as a wave of data collection. Box 1 shows relevant parts of GenV’s Parent/Guardian Information and Consent Form that allows for subsequent contact regarding future invitations to participate in registries, and possible consent wording that registries alongside GenV can use to support optimal data flows.

Box 1Example Parent/Guardian Information and Consent Form (PICF) wording for (a) GenV to work with registries and (b) registries to work with GenV.(a) Wording in the GenV PICF that is specific to supporting registries:“GenV may offer you the chance to take part in future ethically approved studies working with GenV…..You can always choose whether to take part.”“GenV’s data can only be used for ethically approved research to improve health, development, or wellbeing for children and adults. Over time, researchers will use lots of different methods to answer new and important questions. Therefore, the value of your information will keep growing for many years.”“GenV participants may also join studies or registries about specific issues such as head injuries or hearing loss.”“Trials or studies may ask your consent to share data with GenV, with ethics approval. We support this.”(b) Suggested wording for Model 2 registries to include in their PICF, as appropriate to their degree of integration with GenV (Figure 3):“This registry is working with the GenV (Generation Victoria) program. GenV is a research program open to all children living in Victoria and born over two years starting in 2021, and their parents. People in GenV can also be in registries that help prevent, predict and treat problems. This cuts down cost, effort and duplication. It also increases the value of registries. For example, by working with GenV, a registry can access richer data to help solve issues. You can read more about GenV here, and about registries working with GenV here.”“We ask that you consent to allow your registry data to be joined up with your GenV data. Then, both studies can answer more questions about health and other outcomes. Under strict conditions and ethical approval, data from this registry can enter GenV’s dataset, and data from GenV can enter this registry’s dataset.”[Model 2c only] “It is possible that [you/your child] [are/is] eligible for GenV but not enrolled in it. We encourage you to enroll in GenV. This increases the value of this registry, without adding to your time. You can join GenV by… [registry enrolls participant into GenV; registry passes contact details to GenV; parent contacts GenV directly].[Model 2c only—choose relevant wording] You can be in this registry and not in GenV, but we will be missing some information about you/your child *or* you can only take part in this registry if you are also in GenV.”

### 2.5. Measures

GenV may inform registries via socioeconomic or risk factor information, primary or secondary outcomes, eligibility criteria, and moderator and mediator variables. As shown in our Victorian Child’s LifeCourse Journey in Data [15], we propose to link to multiple administrative datasets. Regarding universal biological samples, we are partnering with all of Victoria’s pathology providers to store residual first, second, and third trimester pregnancy samples and newborn blood spots for those who consent, and will collect parent and child saliva soon after birth. Phenotypic measures are under development [16]. GenV’s website will publicly record all datasets accessed each calendar year.

GenV’s lifecourse [17] and outcomes [18] frameworks reflect its stated focus on solutions to improve outcomes and reduce the burden of disease, including via registry and other trials. Therefore, GenV will repeatedly capture overarching health and wellbeing measures comprising health-related quality of life and diagnoses enabling calculation of disease/disability burden [19] related to the International Classification of Diseases [20] and functional status [21].These measures will be coupled with service-related data (e.g., encounters, costs, medications) for economic analyses. GenV is also prioritising outcomes common to multiple Core Outcome Sets [22], rigorously selected for relevance to patients, families, clinicians and policymakers as well as researchers. These span physical, mental, social, cognitive and learning outcomes [18].

Funding permitting, GenV’s biosamples will span multiple tissues (e.g., blood, saliva, stool, breast milk), all participants, and multiple time points including all trimesters of pregnancy, the neonatal period and school entry. Their depletable nature and likely small volume preclude registry-specific assays, with GenV instead prioritising biological data of the broadest value such as metabolomics or polygenic risk data.

Registries will be encouraged to collect a brief common standardised minimum dataset, following integrative research precedents internationally. To be developed collaboratively by 2021, these will likely comprise short demographic and quality of life and PROMS. Benefits include the ability to compare diverse conditions on common metrics and to understand the impacts of co-morbid conditions where children may be in more than one registry.

### 2.6. Data Sharing Considerations

GenV data availability: Like other cohorts, GenV will generally handle data processing *en masse* after each data collection wave/sweep. Given the two-year age span, a data wave would be available much sooner for the youngest than the oldest child participants, and for straightforward survey items than for items requiring extensive processing (e.g., image extraction). Therefore, registries should consider the likely timing of data and its implications.

Data sharing: Figure 4 shows the potential benefits of two-way data sharing, while Figure 5 outlines conceptually the mechanisms by which this might occur under different circumstances. For registries conducted wholly within GenV, all data will initially be within GenV’s Data Repository. For registries conducted alongside GenV, data will need to be shared between the registry and GenV, with appropriate ethics and research agreements adhering to the Five Safes Framework [23] and jurisdictional legislation processes. Data linkage will enable data sharing between GenV and registries that may include aggregated data sets that are established external to both GenV and the registry. When GenV or a registry falls short of target recruitment, one data custodian could periodically approach the other to ascertain whether ‘missed’ persons could be recruited from the other cohort. An important aim of efficient data collection is ‘collect once, use multiple times’. Where duplication of collection of some patient data is anticipated, a Data Sharing Agreement will propose that one of the collaborating partners collects the required data on behalf of both parties and shares it with the other party, in order to reduce overall patient and clinician data collection burden. Whenever possible, deterministic linking of records across different datasets/registries will be employed. In the absence of reliable unique identifiers, probabilistic linking of records across different datasets/registries will be utilised.

By GenV and registries working in partnership, registries can access additional outcomes over more extended time frames and examine variation in response by moderators (such as pre-existing prospectively collected biological or psychosocial traits) or mediators. Registries will also be able to model causal effects (e.g., of having a condition) to the whole population and compare health services and outcomes across registries and for participants with co-morbid conditions. A further benefit is that other researchers can access these ‘depth’ condition-specific data following the period of exclusive registry access (likely six months, as per the UK Biobank and Longitudinal Study of Australian Children).

### 2.7. Governance and Consumer/Stakeholder Considerations

Throughout the period of partnership, GenV will collect agreed data while the registry will maintain independent quality, ethical and governance protocols in line with international standards. The GenV and registry teams will work to understand, prevent and solve any day-to-day issues at either end. The GenV Registries Working Group and GenV staff will work together to fully enable registries as envisaged via enacted governance, funding, and planning activities.

Clear governance arrangements will underpin all collaborations. Model 2 registries will require technical agreements outlining the step-by-step processes for data linkage and transfer. More formal collaborative agreements may or may not be needed initially for registries wholly within (Model 1) or wholly alongside GenV (Model 2c), but collaborations where a registry and GenV seek access to the other’s cohort are more complex. This requires co-ordination (ideally taking a participant perspective) to minimise participant and clinician burden, and documentation by both projects.

The detail of operationalising the integration of GenV and registry-type research remains to be finalised, and is likely to require resourcing both from GenV and the registries themselves—noting that sustained funding is often what makes or breaks any registry in fulfilling its core purposes. Registries are expected to provide sufficient funding to enable start-up, the costs of which will be minimised by GenV’s in-kind provision of core processes and staff expertise. GenV is developing data governance principles that will guide partnership with commercial organisations, including assurance that commercial support will adhere to the same principles and requirements as non-commercially funded research. GenV will use vigilance in limiting red tape and time required to reach the right decisions (e.g., goodness of fit to proceed). As it will be possible for participants to participate in more than one registry, GenV will develop a way to monitor and prevent participant fatigue and not to overburden or compromise either GenV or the participating registries.

## 3. Discussion

To our knowledge, GenV is the first very large cohort internationally aiming to maximise its impact via pre-designed processes for integration with registries from its earliest days. This statement of intent lays out a transparent framework as a roadmap towards operationalisation. It summarises our methodologic developments to (1) catalyse new quality registries coinciding with GenV; (2) enrich existing registries with depth pre- and post-morbid data for its two-year birth window; (3) enable registry-like research within GenV ‘depth’ subcohorts; and (4) enable registries to compare attributes of their condition group with those in other registries, and with the unaffected population, before, during and after its onset.

If we can achieve high population uptake, this would create a systematic route to multiple childhood condition registries, increasing opportunities to answer research questions beyond what a mega-cohort itself can support as depth- or sub-studies. The promise of this approach has already prompted collaborators to develop and launch a new high-risk pregnancy Clinical Quality Registry (Newborn Obstetrics Network Australasia, NONA) alongside GenV (Box 2) and a new special care nursery registry/depth cohort within GenV, aiming to become a stand-alone registry by the end of its first two years of operation. By linking to coronavirus (COVID19) notifications, GenV can also provide a unique whole-state mechanism to track the individual and societal impacts of the pandemic.

Box 2Illustrative example of a new Clinical Quality Registry alongside GenV—the Newborn Obstetrics Network Australasia (NONA).

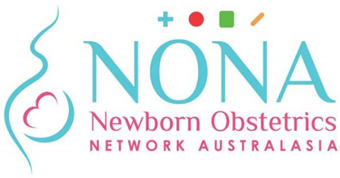

NONA (named after the Roman goddess of pregnancy) is designed to identify the 10% of pregnancies complicated by specific conditions known to be associated with increased perinatal and childhood morbidity and mortality.NONA is a partnership catalysed by GenV between the five Victorian Maternal Fetal Medicine Services (Joan Kirner Women’s and Children’s, Royal Women’s Hospital, Mercy Hospital for Women, Monash Medical Centre, Northern Hospital), Murdoch Children’s Research Institute and Monash Registries.The registry’s establishment will follow principles defined by the Australian Commission on Safety and Quality in Healthcare in conjunction with Monash Clinical Registries. It will be piloted at Joan Kirner Women’s & Children’s with the GenV Vanguard Cohort, and then expand across the other four units.Partnership with GenV means that, for high-risk pregnancies completed in 2021–2023, NONA can access GenV’s (1) universal pregnancy and birth biosamples and (2) long-term phenotypic and health economic outcomes that are otherwise beyond the reach of a clinical registry.The registry will initially focus on three key clinical domains: fetal growth restriction, multiple pregnancies and congenital anomalies with recruitment of participants undertaken within the maternity services either prior to birth or immediately after birth prior to discharge. Opt-out consent into the registry will include consent for linkage to GenV for those participants who consent to participate in GenV. A range of quality metrics have been identified for each clinical domain.This enriched cohort will support discovery in how and why long-term outcomes (not just delivery and survival metrics) may vary by individual, by condition and by service provided.

Strengths & limitations: The GenV design has many strengths relevant to registries. These include a whole-population sampling frame, very large size, reach into regional and rural communities, outcome measures as well as data linkage capacity, which provides a mechanism for long term outcomes beyond those chronicled in medical records and valued by children and families themselves. Prospective pre-morbid health and development pathways—typically a ‘black box’ for registries—may provide new knowledge about risk, resilience, prediction, and disease causation. Multiple registries can be embedded to understand the impacts of multimorbidity and of conditions relative to other diseases and to the rest of the general population. There can be mutual and additive benefits to multiple registries. For example, linkage of antenatal data from NONA to multiple childhood registries may help identify the causal mechanisms of childhood conditions. Such benefits have already been illustrated by the Aspirin in Reducing Events in the Elderly study’s linkage to the Australian Resuscitation Outcomes Consortium Out-of-Hospital Cardiac Arrest registry, enabling the rare outcome of a cardiac arrest to be measured in relation to aspirin use [24].

Regarding limitations, cohort recruitment rates are always unknown at the outset and might not achieve a population denominator. This can partly be mitigated through population-based registries supporting participants who did not initially enter the cohort to do so later, generating both retrospective data linkage and prospective data for the completely ascertained condition group. GenV itself can provide at most an enriched two-year sub-cohort for registries with ongoing recruitment, and its sample size will support uncommon but not rare disorders. It will be vital to minimise the combined burden of the cohort and each registry on participants so that that the goals of both are not compromised. In addition, GenV has purposefully engaged extensively with the public in recognition of the importance of the “social licence” needed for family participation and general community acceptance of a project of this size. This includes social awareness campaigns, peak body endorsement, specific engagement with cultural groups including First Nation groups and concurrent partnership with state government, all of the state’s maternity hospitals and pathology providers. GenV will use its pilot phase to inform specific engagement strategies to maximise recruitment.

Next steps: The next steps are to test how these methods work in practice by operationalising them within GenV. For example, we will learn from and adapt existing governance frameworks and linkage metholodogies, such as the US-based California Health and Human Services Agency frameworks, to be used in the Australian context. These include defining governance arrangements, developing practical processes and templates (e.g., data sharing agreements), and embedding registry capabilities within GenV’s technical platform architecture—all while limiting red tape and ensuring ‘goodness of fit’. We will work towards facilitating registry harmonisation by developing a minimum common dataset, thereby reducing resource duplication and participant burden. GenV can also provide a mechanism to attract sustainable registry funding, once ‘proof of concept’ is established.

## 4. Conclusions

The ability to collect population data once for multiple purposes is of enormous value to epidemiology and to clinical practice. If we can show that a child and parent mega-cohort can support multiple registries then this could answer myriad new questions that are currently either impossible or very slow to answer, while reducing participant burden and ongoing costs of data collection.

## Figures and Tables

**Figure 1 children-08-00285-f001:**
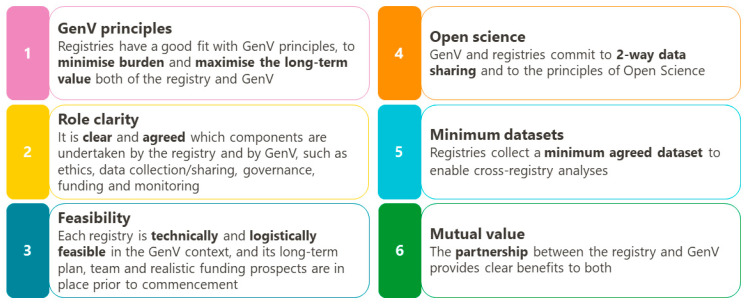
Principles for registries collaborating with GenV.

**Figure 2 children-08-00285-f002:**
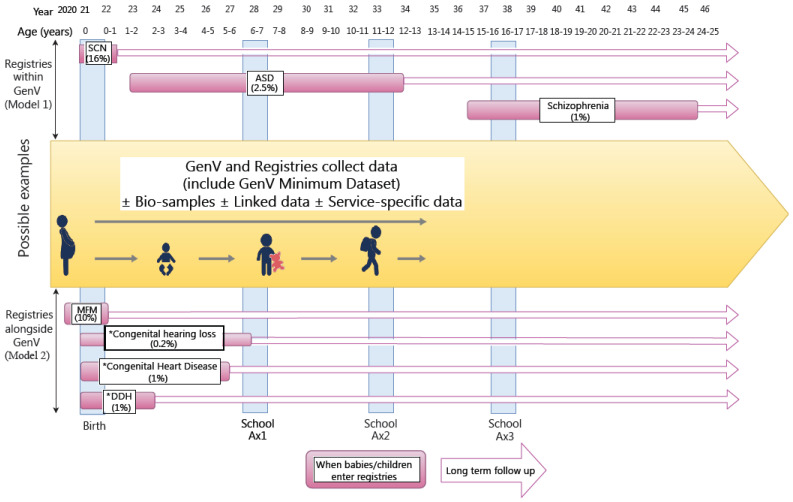
Conceptual relationship of hypothetical registries/depth sub-cohorts to children born in the GenV window. SCN = special care nursery; ASD = autism spectrum disorder; MFM = maternal fetal medicine; DDH = developmental dysplasia of the hip; Ax = assessment. * = existing registries. Refer to Figure 3 regarding Models 1 & 2.

**Figure 3 children-08-00285-f003:**
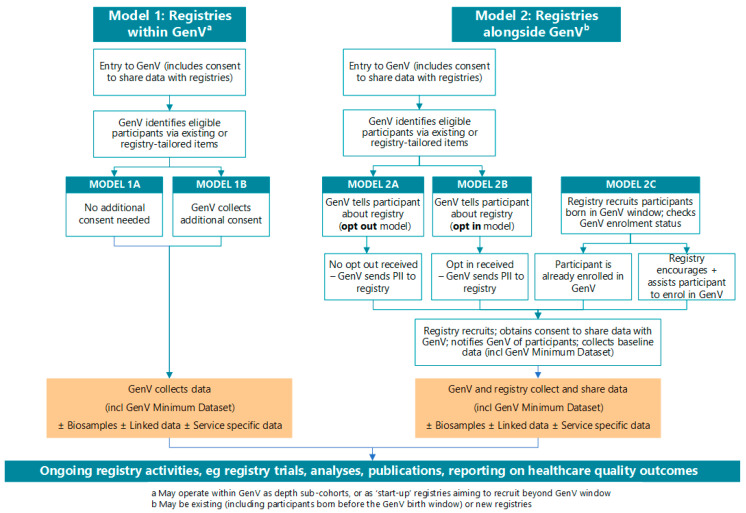
Process flowchart for registries within and alongside GenV. Note: further information on data sharing is captured in Figure 5. PII = personal identifying information required for registry to approach participants to invite into registry.

**Figure 4 children-08-00285-f004:**
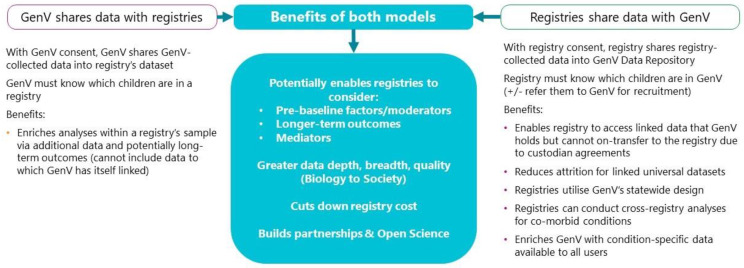
Benefits of registries sharing data with GenV and of GenV sharing data with registries. Applies to children born in the GenV window (2021–2023) and their parents.

**Figure 5 children-08-00285-f005:**
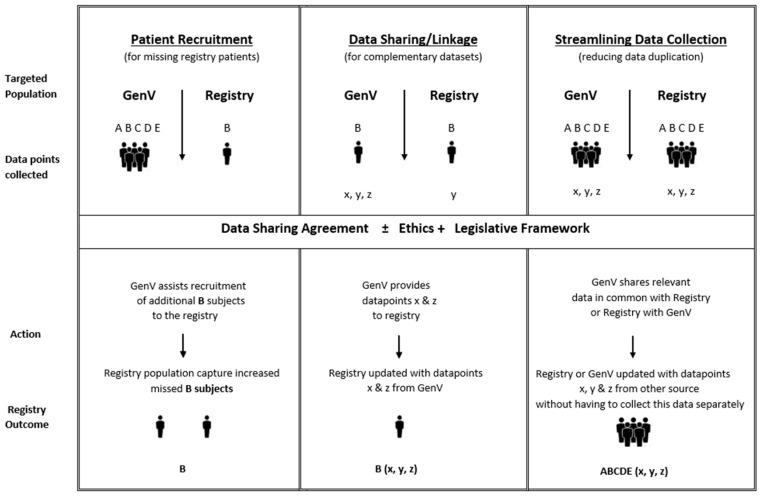
GenV data sharing models. This figure illustrates flow from GenV to registries. The reverse is also expected; that is, the figure can be mirrored to illustrate flow from registries to GenV.

## Data Availability

No new data were created or analysed in this study. Data sharing is not applicable to this article. In the future, GenV proposes to disseminate on-the-spot individual results that may have the potential to improve parent or child health and wellbeing, and relevant summary GenV findings through preferred channels, and GenV data access will be supported for all researchers meeting governance requirements such as the Five Safes principles. A range of materials are available at GenV’s figshare project [https://mcri.figshare.com/projects/Generation_Victoria/35822 (accessed on 2 April 2021)].

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
