# Peer review of "Enhancing Value and Uptake for Whole-Population Cohorts of Children and Parents: Methods to Integrate Registries into the Generation Victoria Cohort"

_children, 2021, doi:10.3390/children8040285_

Round 1

Reviewer 1 Report

This is an interesting paper that outlines principles, methods and governance to integrate registries alongside a mother-baby cohort. This cohort and proposed linkage to registries is a novel project and these methods will be of use for future projects in other jurisdictions.  I just have a few questions:

Consent:

In Model 2, will participants be asked to consent each time a new registry is to be added? Won’t this add to burden of participation?

Figure 2:

On Figure 2 – add which ones are included in Model 1 & Model 2 so that it can be cross-referenced to other figures. Even better would be examples of Models 1a & b, 2 a, b, & c.

Figure 5:

Figure 5 is confusing. With the duplication of data collection – if the registry is already collecting data, why would GenV collect the same information?  Isn’t GenV only running for 2 years – I would think that the registry would provide the data rather than GenV.

Reviewer 2 Report

I greatly appreciated this piece. It describes an innovative model for collecting and analyzing population-level data, one that hold significant promise for generating new knowledge and improving health for all. A few areas that I would appreciate more information:

  1. Linkage methodology: Is the linkage deterministic or will there be probabilistic elements? More information about which methodology would be employed in which cases would be helpful.
  2. Cultural License: Is the public ready for this type of research? And who would be running engagement efforts? Its success hinges on engagement, because without widespread engagement, the denominator isn't an entire population. 
  3. Funding / Staffing: Although there is a section that describes funding, it does little to describe how initial start-up would be funded and carried out, which comprises a huge share of the work and requires completely different skill sets from the subsequent efforts. 
  4. Although the article is understandably focused on gleaning lessons learned from somewhat similar efforts within Australia, it does not reference more similar efforts in the US and elsewhere. Specifically, work in California could inform the governance framework and linkage methodology, as well as other parts of the work (CHHS Data Sharing Framework; Integrating Data to Advance Research, Operations, and Client-Centered Services in California).
